# Association of Maternal Dietary Patterns during Pregnancy and Offspring Weight Status across Infancy: Results from a Prospective Birth Cohort in China

**DOI:** 10.3390/nu13062040

**Published:** 2021-06-15

**Authors:** Jiajin Hu, Izzuddin M. Aris, Pi-I D. Lin, Ningyu Wan, Yilin Liu, Yinuo Wang, Deliang Wen

**Affiliations:** 1Health Sciences Institute, China Medical University, Shenyang 110122, China; jjhu@cmu.edu.cn (J.H.); WNYjudy@163.com (N.W.); cmuylliu@163.com (Y.L.); ynwang@cmu.edu.cn (Y.W.); 2Research Center of China Medical University Birth Cohort, China Medical University, Shenyang 110122, China; 3Division of Chronic Disease Research across the Lifecourse, Department of Population Medicine, Harvard Medical School, Boston, MA 02215, USA; Izzuddin_Aris@harvardpilgrim.org (I.M.A.); pil864@mail.harvard.edu (P.-I.D.L.)

**Keywords:** pregnancy, maternal nutrition, dietary patterns, infant, weight status, obesity

## Abstract

Literature on maternal dietary patterns during pregnancy and offspring weight status have been largely equivocal. We aimed to investigate the association of maternal dietary patterns with infant weight status among 937 mother–infant dyads in a Chinese birth cohort. We assessed maternal diet during pregnancy using food frequency questionnaires (FFQ) and three-day food diaries (TFD) and examined infants’ body weight and length at birth, 1, 3, 6, 8 and 12 months. Maternal adherence to the “protein-rich pattern (FFQ)” was associated with lower infant body mass index z-scores (BMIZ) at birth, 3 and 6 months and lower odds of overweight and obesity (OwOb) across infancy (quartile 3 (Q3) vs. quartile 1 (Q1): odds ratio (OR): 0.50, (95% confidence interval: 0.27, 0.93)). Maternal adherence to the “vegetable–fruit–rice pattern (FFQ)” was associated with higher BMIZ at birth, 3 and 6 months and higher odds of OwOb across infancy (Q3 vs. Q1: OR: 1.79, (1.03, 3.12)). Maternal adherence to the “fried food–bean–dairy pattern (TFD)” was associated with lower BMIZ at 3, 6, 8 and 12 months and lower odds of OwOb (Q3 vs. Q1: OR: 0.54, (0.31, 0.95)). The study results may help to develop interventions and to better define target populations for childhood obesity prevention.

## 1. Introduction

Childhood obesity remains a global public health concern [1,2,3], as childhood obesity often tracks into adulthood and is strongly associated with adult chronic diseases such as diabetes and metabolic syndrome [4,5]. Previous studies indicated that accelerated growth in early life, especially during infancy, is a strong risk factor for subsequent childhood obesity [6,7]. Thus, identifying modifiable determinants of infant growth may provide insight into intervention strategies for childhood obesity prevention in early life.

The recent report from the Commission of Ending Childhood Obesity [8] highlighted the importance of prenatal care, such as appropriate maternal nutrition during pregnancy, as a key modifiable factor and strategy in childhood obesity prevention. While several epidemiological studies have examined associations of maternal dietary patterns with offspring weight status, they did not show consistent results [9,10,11,12,13]. In the Generation R study in Netherlands [9], Broek et al. derived three food patterns: a vegetable–fish oil pattern, a nuts–soy–high-fiber cereals pattern and a margarine–snacks–sugar dietary pattern; however, none of these patterns were associated with children’s body mass index (BMI) or fat mass index at 6 years of age. However, in a cohort study in the United States, Martin et al. [11] reported that a maternal dietary pattern characterized by high intake of white bread, meats and fried foods was associated with higher age-specific BMI z-scores at 1 and 3 years of age. A systematic review [14] indicated that mothers who adhered to a Mediterranean diet during pregnancy gave birth to children who were at lower risk of being overweight and had a more beneficial profile of metabolic markers such as glycaemia and blood pressure. However, most of these studies were based in Western countries, and only one study was based in Asia [12]. In the Growing Up in Singapore Towards healthy Outcomes study in Singapore, Chen et al. [12] reported that a vegetable–fruit–white rice dietary pattern during pregnancy was associated with lower offspring adiposity. To our knowledge, no previous studies have examined the association between maternal dietary patterns and offspring weight status in China, a developing country currently experiencing a rapid transition of dietary habits and also a rapid increase in prevalence of childhood obesity [1,15]. Besides, few studies have assessed maternal dietary patterns with infant growth trajectories, an important risk marker for later-life obesity development [16].

Moreover, previous studies have mostly used a single dietary assessment tool to assess dietary intake, such as food frequency questionnaires (FFQ) [9,13] or food records [12]. However, both tools have their own limitations [17,18]. Although FFQs can capture an individual’s long-term habitual diet, it often overestimates food consumption and is susceptible to recall bias [19,20]. While food dairies may provide more accurate assessments of intake of specific food items, they usually do not reflect stable and long-term food intake [17]. The combined usage of the two tools will provide a more comprehensive assessment of diet.

To address these research gaps, we aimed to investigate the association of maternal dietary pattern with infant weight status across infancy using a large prospective pre-birth cohort in China. We hypothesized that maternal dietary patterns may predict infant growth trajectories and healthy dietary patterns during pregnancy were associated with decreased risk of overweight and obesity (OwOb) across infancy.

## 2. Materials and Methods

### 2.1. Study Population and Design

We used data from the “Born in Shenyang Cohort Study” (BISCS), a prospective pre-birth cohort in the northeastern region of China. The study design has been described elsewhere [21]. Briefly, between April and September 2017, we enrolled women with singleton pregnancies between 21 and 24 gestational weeks from 54 community healthcare centers and hospitals, which provided antenatal and maternity care in the urban areas of Shenyang. Among 2068 eligible pregnancies, 1338 agreed to participate, 1296 mothers had singleton live births. In the present analysis, we excluded mothers with incomplete dietary assessments during pregnancy (*n* = 85) or implausible values of total calorie intake (<500 or >3500 kcal/day) [12] (*n* = 138) and infants with no anthropometric data at all 5 follow-up visits after birth (*n* = 136), leaving 937 mother–infant pairs in the analysis. We compared characteristics of pregnancies included in the present analysis (*n* = 937) with pregnancies excluded (*n* = 359) and found that the mothers who were excluded tended to be multiparous, nonsmokers, had older age, less likely to be of Han Chinese ethnicity, had high calorie intake level and lower physical activity level (Appendix A).

Trained research staff conducted in-person interviews at the enrollment visit during middle pregnancy (mean gestational age 22 weeks, standard deviation (SD) 1.2), and conducted follow-up visits to mothers and infants at birth, 1 month (response rate: 72.4%), 3 months (71.2%), 6 months (66.7%), 8 months (77.8%) and 12 months (77.4%) after birth. All women gave written informed consent to participate. The ethics committee of the China Medical University approved the study.

### 2.2. Exposures

We assessed women’s prenatal diet during pregnancy using both three-day food diaries (TFD) and FFQ. At the enrollment visit, we asked pregnant women to complete a TFD over two consecutive weekdays and one weekend day. We trained pregnant women to record their food and drink consumption, with a detailed list of all ingredients and their portions (in grams). We provided participants with visual-aid booklets with colored photographs of 200 local food items with standard portion sizes to help them better identify the type and amount of food. We collected the handwritten TFDs before 24 weeks of gestation and summarized all food items into 21 non-overlapping food groups. We averaged the daily consumption for each food group over three days and derived daily calorie and nutrient intakes according to the China Food Composition Database [22]. We also asked participants to report their dietary intake frequency from pregnancy to enrollment visit (mean gestational age 22 weeks, SD 1.2) using a semiquantitative FFQ. The FFQ contained 25 food items and beverages with nine consumption frequency categories: “almost never”, “<1 time per month”, “1–3 times per month”, “1–2 times per week”, ”3–4 times per week”, “5–6 times per week”, “1 time per day”, “2 time per day” and “≥3 times per day”. We did not collect data on the amount consumed in the FFQ, thus we calculated only the daily consumption frequency for each food group. To be comparable with the TFD, we further combined the 25 FFQ food groups into the 21 TFD food groups based on their similarity in nutrient profiles (e.g., combined pork, beef, and lamb as a red meat group). To evaluate the reproducibility of the FFQ, we invited a subset of 401 pregnancies without gestational diabetes mellitus (GDM) to recomplete the FFQ at the third trimester (mean gestational age 33 weeks, SD 1.1). We chose pregnancies without GDM for the reproducibility study because GDM participants may take doctors’ dietary advice and change their diet habits after the diagnosis. The Spearman correlation coefficients of consumption frequencies ranged from 0.80 (vegetables) to 0.43 (tubers) between the two FFQs, and ranged from 0.60 (vegetables) to 0.25 (whole grains) between FFQ (the first time) and TFD.

### 2.3. Outcomes

We extracted birth weight and length of neonates from medical records. We measured infants’ weight (to the nearest 0.01 kg) and length (to the nearest 0.1 cm) twice and averaged the two measurements during research visits at ages 1, 3, 6, 8 and 12 months with calibrated weighing scales (Seca 376+; Seca Corporation, Hamburg, Germany) and infant stadiometers (Seca 416; Seca Corporation, Hamburg, Germany). We calculated BMI by dividing infant weight (kilograms) by the square of height (meters). We derived age- and sex-specific z-scores for body mass index (BMIZ) and sex-specific weight for length z-scores (WFLZ) based on the World Health Organization (WHO) child growth standards [23]. We defined OwOb as WFLZ ≥2.

### 2.4. Covariates

Pregnant women reported their sociodemographic characteristics such as age at enrollment, ethnicity, education attainment, household income, parity and behaviors during pregnancy such as smoking status and physical activity using interviewer-administered questionnaires. We treated mothers’ age as continuous variables, except for the descriptive statistic, which we categorized into four groups (<25, 25–29, 30–34, ≥35 years). We categorized ethnicity into two groups (Han Chinese vs. Others), mothers’ educational attainment into four groups (middle school or below, high school, college, graduate or above), parity into two groups (primiparous vs. multiparous), and smoking status into two groups (never vs. ever). We assessed mothers’ physical activity level using the Pregnancy Physical Activity Questionnaire (Chinese version) [24], and categorized physical activity levels (metabolic equivalents (MET)) into three groups (<100 MET-hour/week, 100 to <200 MET-hour/week, ≥200 MET-hour/week). We measured the height of pregnant women at the enrollment visit with a calibrated stadiometer (Seca 217; Seca Corporation, Hamburg, Germany). Mothers self-reported their prepregnancy height and fathers self-reported their height and weight at enrollment. We calculated mothers’ and fathers’ BMI by dividing weight (kilograms) by the square of measured height (meters) and categorized maternal and paternal BMI as underweight (BMI < 18.5 kg/m^2^), normal weight (18.5 kg/m^2^ ≤ BMI < 24 kg/m^2^) or OwOb (BMI ≥ 24 kg/m^2^) using WHO references for the Asian population [25]. We derived pregnancies’ daily calorie intake from the TFD and categorized pregnancies’ calorie intake into two groups (<2100 kcal/d vs. ≥2100 kcal/d) according to the recommend calorie intake for women in the second trimester by dietary guidelines for Chinese residents [26]. We defined total gestational weight gain as the difference between last recorded weight before delivery and self-reported prepregnancy weight.

### 2.5. Statistical Analysis

We performed principal component analysis for TFD and FFQ separately to identify dietary patterns. We applied varimax rotation for greater interpretability. We first standardized the food consumption frequency (FFQ) and the daily food consumption level (TFD) into z-scores to reduce the influence of greater variance of food items. Then, we used standardized daily food consumption frequency in the FFQ and daily food intake level in the TFD to identify dietary patterns, separately. Among all factors derived from FFQ and TFD, we identified distinct factors as major dietary patterns based on eigenvalue (>1), factor interpretability after varimax rotation, and a scree plot (factors appeared before significant inflection point of slope) (Appendix A), showing the proportion of the variance of total consumption of the food variables. We calculated dietary pattern scores by summing the standardized food intake level (TFD) or standardized food consumption frequencies (FFQ) by weighted corresponding factor loadings. We identified main contributors to dietary pattern as food items with an absolute factor loading value greater than 0.32, in accordance with a previous study [27]. We performed Spearman correlation analysis to test correlations between dietary patterns derived from the two assessment methods. We further categorized mothers’ dietary pattern scores into quartiles for subsequent analyses.

We compared dietary pattern scores across participants’ social demographic characteristics using *t*-tests or ANOVA tests. We used linear mixed effects (LME) models to examine the longitudinal associations between mother’s dietary pattern scores (in quartiles) during pregnancy and offspring BMIZ across infancy. The LME models take into account within-subject correlation of repeated measurements and allows for incomplete outcome measurement at the same time. We included interaction terms between maternal dietary patterns and child age to estimate the change in BMIZ over time associated with higher maternal dietary patterns scores. We fitted the models using an unstructured covariance matrix for random effects variables (intercept and slope) and used the maximum likelihood estimation method. We estimated adjusted differences (and 95% confidence interval (CI)) in predicted BMIZ at birth, 3, 6, 8 and 12 months between quartiles of dietary pattern scores using the margins command in Stata. In the sensitivity analysis, we also used WFLZ as an indicator for infant weight status to assess the robustness of the results. We used generalized estimating equation models to assess longitudinal associations between mothers’ dietary pattern scores during pregnancy and children’s OwOb risk across infancy. We conducted unadjusted and adjusted analyses: model 1: unadjusted; model 2: adjusted for other dietary patterns derived from the same dietary assessment tool (TFD or FFQ); model 3: adjusted for prepregnancy BMI, age, parity, family income, education level, ethnicity, smoking status, total daily calorie intake, physical activity and paternal BMI; model 4: model 2 + model 3. We further adjusted for total gestational weight gain to examine whether the associations can be explained by maternal weight status. We performed all analyses using Stata S.E. version 13 (Stata Corp., Texas, TX, USA).

## 3. Results

### 3.1. Participant Characteristics

Among 937 pregnant women in our analytic sample, 79.3% were between 25 and 35 years of age at enrollment, 84% were of Han Chinese ethnicity (Table 1), 72.3% of participants have educational attainment of college or above., 78.9% of participants were primiparous and 26.5% were OwOb (BMI ≥ 24 kg/m^2^) before pregnancy. More than 19% of participants were at a higher physical activity level (≥200 MET-hour/week) and 38.5% of participants had a higher level of calorie intake (≥2100 Kcal/d), 59.5% of the fathers were OwOb, 52.3% of the infants were female and the average infants’ BMIZ increased from birth (−0.36 ± 1.14, SD units) to 12 months after birth (0.53 ± 0.95, SD units).

### 3.2. Dietary Patterns

Among six factors derived from the FFQ, we identified three dietary patterns that accounted for 32.27% of the total variation (Table 2). The first pattern included higher intakes of marine fish, shrimps, crabs and mussels, freshwater fish, seaweed and haslet (i.e., “fish–seafood” pattern). The second dietary pattern included higher intakes of proteins such as dairy products, milk, eggs and products, beans and products and nuts (i.e., “protein-rich” pattern). The third pattern included higher intakes of vegetables, fruits and rice (i.e., “vegetable–fruit–rice” pattern).

Among nine factors derived from the TFD, we identified four dietary patterns that accounted for 26.31% of the total variation (Table 2). The first pattern, which we named “traditional pattern (TFD)”, included higher intakes of tubers, vegetables, fruits, red meat and rice. The second pattern included higher intakes of pastry, candy, sweet beverages, shrimps, crabs, and mussels (i.e., “sweet foods” pattern). The third pattern included higher intakes of fried foods, beans and products and dairy products (i.e., “fried food–beans–dairy” pattern). The fourth pattern included higher intakes of coarse grains, shrimps, crabs and mussels and a low intake of eggs and rice (i.e., “whole grain–seafood” pattern). We observed a significant positive correlation between “vegetable–fruit–rice pattern (FFQ)” scores and “traditional pattern (TFD)” scores (*p* < 0.001), (Appendix A).

Table 3 describes the dietary pattern scores according to characteristic groups of the participants. The dietary pattern scores were distributed from negative numbers to positive numbers. Generally, we observed that maternal age, educational attainment, parity, prepregnancy BMI, physical activity and total daily calorie intake level during pregnancy and household income were associated with at least one dietary pattern score during pregnancy (Table 3).

### 3.3. Associations of Maternal Dietary Patterns and Infant Weight Status

After adjusting for confounders, we observed that maternal adherence to “protein-rich pattern (FFQ)” was associated with lower BMIZ of infants at birth, 3 and 6 months (Table 4). Exposure to higher maternal “protein-rich pattern (FFQ)” score quartiles was associated with lower odds of OwOb across infancy (quartile 3 (Q3) vs. quartile 1 (Q1): odds ratio (OR): 0.50, (95% CI: 0.27, 0.93)) (Table 5). Maternal adherence to the “vegetable–fruit–rice pattern (FFQ)” was associated with higher infant BMIZ at birth, 3 and 6 months (Table 4). Exposure to higher maternal “vegetable–fruit–rice pattern (FFQ)” score quartiles was associated with higher odds of OwOb across infancy (Q3 vs. Q1: OR: 1.79, (1.03, 3.12)) (Table 5). Maternal adherence to the “fried food–bean–dairy pattern (TFD)” was associated with lower infant BMIZ at 3, 6, 8 and 12 months (Table 4), and exposure to higher maternal “fried food–bean–dairy pattern (TFD)” score quartiles was associated with lower odds of OwOb across infancy (Q3 vs. Q1: OR: 0.54, (0.31, 0.95)) (Table 5). In sensitivity analyses, results remained stable when using WFLZ as an indicator of infant weight status instead of BMIZ (Appendix A). Further, adjusting for maternal gestational weight gain generally did not change the results (Appendix A). We also observed higher “protein-rich pattern (FFQ)” score quartiles and “fried food–bean–dairy pattern (TFD)” score quartiles were associated with lower infant BMIZ (Appendix A) and WFLZ (Appendix A) across infancy. Fish–seafood pattern (FFQ) scores, traditional pattern (TFD) scores and sweet foods pattern (TFD) scores were not associated with infant BMIZ or OwOb risk. The interaction items between maternal dietary patterns and infant age were not significant (Appendix A).

## 4. Discussion

In this prospective birth cohort in China, we observed some evidence that maternal adherence to dietary patterns characterized by higher protein intake (i.e., “protein-rich pattern (FFQ)” and “fried food–bean–dairy pattern (TFD)”) during pregnancy were associated with lower infant BMI z-score and lower likelihood of being OwOb across infancy. Maternal adherence to dietary patterns characterized by higher vegetable, fruit and rice intake (i.e., “vegetable–fruit–rice pattern (FFQ)”), however, was associated with an increased risk of being OwOb and higher infant BMI z-score.

Our findings regarding adherence to protein-rich dietary patterns is partly consistent with previous studies showing that maternal protein intake during pregnancy was inversely associated with infants’ OwOb risk [28]. Although our “fried food–bean–dairy pattern (TFD)” contains fried food, which is believed to be a risk factor for obesity development [29], the pattern also contains high-protein foods such as beans and dairy. Besides, the fried food in the study also contained plenty of high-protein foods like fried fish or meat. A previous review [30] showed that both low and high maternal dietary protein intake was associated with intrauterine growth restriction, which is a known risk marker for metabolic disorders, abnormal development, and cardiovascular disorders in later life [31]. The mechanisms of maternal intrauterine nutrition programming offspring obesity are not yet fully understood. Possible mechanisms could be that maternal protein nutrition has a pronounced impact on fetal programming and alters the expression of genes in the fetal genome, thus leading to metabolic disorders, organ dysfunction, hormone imbalances, and cell signaling defects at childhood [30]. Further, mothers with higher scores for the “protein-rich pattern (FFQ)” were of younger age and had higher educational attainment, both of which have been shown to be inversely associated with risk of offspring obesity [32]. However, these associations remained after adjusting for social socioeconomic and demographic characteristics and other confounders. We did not observe a linear dose–response relationship between OwOb risk and the two patterns, which suggests that there may be optimal threshold value of the patterns.

Few studies have examined the associations of dietary patterns characterized by higher intakes of vegetables, fruits and rice with offspring weight status. In a previous study in Singapore [12], Chen et al. reported that the vegetable–fruit–white rice pattern was associated with lower postnatal offspring BMI z-score, which is inconsistent with our results. However, in that study, the “vegetable–fruit–white rice” pattern was found to be associated with offspring weight status at ages 18 and 48 months, but not in infancy. Taken together, our findings suggest that the influence of this specific dietary pattern on child development may vary across time [33]. Tubers, rice and fruits are generally regarded as high-carbohydrate and high-glycemic-index foods, which may influence maternal blood glucose levels and intrauterine nutritional environment, leading to change in offspring metabolic programming [34]. An animal study also indicated that a low-carbohydrate diet with lower glycemic index during gestation may be critical to regulate the programming of adipogenesis in the offspring [35].

In our study, the “whole grain–seafood pattern (TFD)” scores were negatively associated with infant BMI z-score at quartile 2 (Q2), the estimated value remained negative at Q3 and Q4, though not statistically significant. In a previous study, maternal dietary pattern characterized by a higher consumption of whole grains predicted a greater newborn fat-free mass but no difference in fat mass or adiposity [36]. Maternal high adherence to a Mediterranean diet score, which was characterized by high consumption of seafood and cereals, was linked to a lower waist circumference of offspring in early childhood [37].

The strengths of our study include its prospective design, which is important in establishing the temporal sequence between maternal dietary patterns and offspring growth. Our study also fills the knowledge gap on the maternal dietary patterns in a relatively understudied Asian population. Our study is also strengthened by having repeated measures of offspring length and weight status at multiple time points during infancy, which enabled us to observe infant weight status trajectories. We also used multiple dietary assessment instruments to capture maternal diet; the TFD has the advantage in the accurate assessment of short-term food intake, while the FFQ captured longitudinal habitual diet across pregnancy.

Several limitations of our study should be considered. First, the FFQ used in the present study was relatively brief, which was comprised of 25 food item groups. However, these 25 food groups represented the most commonly consumed dishes among the Chinese population and these items showed a high proportion of overlap with the food items recorded in the TFD. Second, the identified dietary patterns included in the present analysis accounted for a relatively small proportion of the total variation (32.27% for the FFQ and 26.31% for the TFD). This observation may be due to the complexity of Chinese dietary patterns and many other combinations of patterns of food consumption were not identified as distinct dietary patterns. Other studies [38,39] focusing on Chinese dietary patterns have similarly reported a small proportion variation of the total intake from FFQs. Third, the study was based in an urban setting with a relatively high socioeconomic status; however, dietary patterns are similar between rural and urban populations in China according to a previous study [40]. Fourth, even though the FFQs collected participants’ dietary habits from early pregnancy, the TFDs were only conducted at middle pregnancy, thus they may not adequately capture maternal diet throughout the pregnancy. However, previous studies reported that dietary patterns across pregnancy are likely to remain stable [41]. We also lacked measurements of the pregnancy dietary habits of mothers, thus we were not able to observe the association between pregnancy diet and offspring weight status. Fifth, our findings may be subject to selection bias because of loss to follow-up, and the fact that women excluded from the analytic cohort were more likely to be multiparous, at older age, had a high calorie intake level and a lower physical activity level, which were associated with maternal dietary patterns. Sixth, though the study was based on a community population, which may provide relatively well generalizability of a local population, our findings are based on a regional population, thus they may not generalize to other settings. Lastly, there may be residual confounding such as infants’ genetic risk of obesity, which we have not accounted for in this analysis.

## 5. Conclusions

In summary, we found several maternal dietary patterns that were associated with offspring weight status across infancy. Dietary patterns characterized by higher protein intake, such as the “protein-rich pattern (FFQ)” and the “fried food–bean–dairy pattern (TFD)”, were associated with lower OwOb risk and lower BMI z-score throughout infancy, while dietary patterns characterized by higher vegetable, fruit and rice intake were associated with higher risk of OwOb and higher BMI z-score throughout infancy. Healthcare providers should pay special attention to infants whose mothers have a high adherence to obesogenic dietary patterns during pregnancy to avoid obesity. Identifying specific dietary patterns in pregnancy associated with infant growth could also inform preventive strategies to reduce child obesity.

## Figures and Tables

**Table 1 nutrients-13-02040-t001:** Characteristics of participants in the “Born in Shenyang” cohort (*n* = 937).

Characteristics	Mean ± SD or *n* (%)
**Mothers**	
Age at enrollment (Years)	
<25	53 (5.7)
25–29	399 (42.6)
30–34	344 (36.7)
≥35	141 (15.1)
Ethnicity	
Han	787 (84.0)
Others	150 (16.0)
Educational attainment	
Middle school or below	76 (8.1)
High school	146 (15.6)
College	628 (67.0)
Graduate or above	87 (9.3)
Household income per year, CNY	
<30,000	247 (26.4)
30,000 to <50,000	256 (27.3)
50,000 to <70,000	210 (22.4)
≥70,000	224 (23.9)
Parity	
Primiparous	739 (78.9)
Multiparous	198 (21.1)
Smoking status	
Yes	4 (0.4)
No	933 (99.6)
Prepregnancy BMI category	
<18.5, kg/m^2^	123 (13.1)
18.5 to <24.0, kg/m^2^	566 (60.4)
24.0 to <28.0, kg/m^2^	186 (19.9)
≥28.0, kg/m^2^	62 (6.6)
Physical Activity	
<100 MET-hour/week	230 (24.6)
100 to <200 MET-hour/week	527 (56.2)
≥200 MET-hour/week	180 (19.2)
Calorie intake	
<2100 kcal/d	576 (61.5)
≥2100 kcal/d	361 (38.5)
**Fathers**	
BMI at enrollment (*n* = 903)	
<18.5, kg/m^2^	22 (2.4)
18.5 to <24.0, kg/m^2^	344 (38.1)
24.0 to <28.0, kg/m^2^	341 (37.8)
≥28.0, kg/m^2^	196 (21.7)
**Infants**	
Sex	
Male	447 (47.7)
Female	490 (52.3)
BMIZ at birth, SD units (*n* = 698)	−0.36 ± 1.14
BMIZ at 1-month visit, SD units (*n* = 781)	0.02 ± 0.95
BMIZ at 3-month visit, SD units (*n* = 772)	0.30 ± 1.04
BMIZ at 6-month visit, SD units (*n* = 724)	0.35 ± 1.05
BMIZ at 8-month visit, SD units (*n* = 817)	0.35 ± 0.99
BMIZ at 12-month visit, SD units (*n* = 819)	0.53 ± 0.95

SD: standard deviation; CNY, Chinese Yuan; MET, metabolic equivalents; BMIZ: age- and sex-specific z-scores for body mass index based on the World Health Organization child growth standards.

**Table 2 nutrients-13-02040-t002:** Rotated factor loadings of the distinct dietary patterns derived from the FFQ and the TFD.

Dietary Patterns	Food	Factor Loading Coefficient	Variance Explained (%)
Food Frequency Questionnaire			
Fish–seafood pattern (FFQ)			13.12
	Marine fish	0.83	
	Shrimps, crabs and mussels	0.79	
	Freshwater fish	0.77	
	Seaweed	0.62	
	Haslet	0.54	
Protein-rich pattern (FFQ)			10.32
	Dairy products	0.73	
	Milk	0.73	
	Eggs and products	0.64	
	Beans and products	0.54	
	Nuts	0.45	
Vegetable–fruit–rice pattern (FFQ)			8.83
	Vegetables	0.83	
	Fruits	0.81	
	Rice	0.57	
Three-Day Food Dairies			
Traditional pattern (TFD)			7.83
	Tubers	0.73	
	Vegetables	0.57	
	Fruits	0.53	
	Red meat	0.41	
	Rice	0.40	
Sweet foods pattern (TFD)			6.30
	Pastry and candy	0.67	
	Sweet Beverages	0.66	
	Shrimps, crabs and mussels	0.42	
Fried food–bean–dairy pattern (TFD)			6.12
	Fried foods	0.75	
	Beans and products	0.66	
	Dairy products	0.36	
Whole grain–seafood pattern (TFD)			6.06
	Coarse grains	0.67	
	Shrimps, crabs and mussels	0.34	
	Eggs	−0.35	
	Rice	−0.48	

FFQ: food frequency questionnaire; TFD: three-day food diaries.

**Table 3 nutrients-13-02040-t003:** Dietary pattern scores according to participants’ characteristics among 937 mother–child pairs in the “Born in Shenyang” cohort.

Characteristics	Dietary Pattern Scores, Mean (SD)						
	Food Frequency Questionnaire			Three-Day Food Dairies			
	Fish–Seafood Pattern (FFQ)	Protein-Rich Pattern (FFQ)	Vegetable–Fruit–Rice Pattern (FFQ)	Traditional Pattern (TFD)	Sweet Foods Pattern (TFD)	Fried Food–Bean–Dairy Pattern (TFD)	Whole Grain–Seafood Pattern (TFD)
Maternal							
Age at enrollment (Years)							
<25	−0.56 (1.83)	−0.56 (2.31)	−0.37 (1.73)	0.46 (1.63)	0.23 (1.67)	0.16 (1.91)	−0.27 (1.21)
25–29	0.18 (3.43)	0.30 (2.63)	0.15 (2.06)	0.11 (1.76)	0.05 (1.40)	0.04 (1.31)	−0.11 (1.14)
30–34	−0.10 (2.58)	−0.17 (2.17)	−0.12 (2.04)	−0.10 (1.61)	−0.04 (1.16)	0.09 (1.27)	0.11 (1.46)
≥35	0.03 (2.79)	−0.20 (2.33)	0.08 (2.02)	−0.12 (1.79)	−0.11 (1.49)	−0.21 (1.18)	0.13 (1.34)
*p*	0.297	0.009	0.136	0.066	0.358	0.115	0.030
Ethnicity							
Han	0.05 (2.97)	−0.19 (2.37)	−0.01 (2.01)	0.00 (1.70)	0.01 (1.34)	0.04 (1.34)	0.02 (1.29)
Others	−0.19 (2.96)	0.11 (2.65)	0.12 (2.13)	0.10 (1.74)	−0.01 (1.39)	−0.02 (1.21)	−0.11 (1.39)
*p*	0.354	0.553	0.480	0.488	0.917	0.624	0.264
Educational attainment							
Middle school or below	−0.60 (2.47)	−0.89 (2.32)	−0.18 (2.19)	0.29 (1.66)	−0.26 (1.02)	−0.10 (1.27)	−0.27 (1.01)
High school	−0.17 (2.77)	0.01 (2.46)	−0.10 (2.15)	0.17 (1.76)	−0.12 (1.19)	−0.09 (1.19)	0.00 (1.16)
College	0.10 (3.08)	0.04 (2.49)	0.03 (2.01)	0.00 (1.71)	0.07 (1.35)	0.08 (1.37)	−0.05 (1.30)
Graduate or above	0.23 (2.87)	0.48 (1.86)	0.20 (1.86)	−0.33 (1.56)	−0.01 (1.77)	−0.02 (1.20)	0.55 (1.62)
*p*	0.191	0.003	0.585	0.081	0.132	0.390	<0.001
Household income per year, CNY							
<30,000	−0.14 (2.65)	−0.13 (2.51)	−0.22 (1.97)	0.07 (1.78)	−0.16 (1.14)	−0.11 (1.12)	−0.10 (1.17)
30,000 to <50,000	−0.21 (2.27)	−0.16 (2.08)	−0.02 (1.96)	0.12 (1.70)	−0.04 (1.10)	0.11 (1.44)	−0.09 (1.20)
50,000 to <70,000	0.04 (2.82)	0.05 (2.64)	−0.05 (2.02)	0.00 (1.60)	0.12 (1.62)	0.06 (1.38)	−0.19 (1.31)
≥70,000	0.42 (3.96)	0.29 (2.43)	0.24 (2.18)	−0.14 (1.73)	0.12 (1.62)	0.07 (1.33)	0.21 (1.52)
*p*	0.096	0.166	0.018	0.387	0.067	0.254	0.042
Parity							
primiparous	0.07 (3.07)	0.13 (2.50)	0.00 (2.01)	0.01 (1.70)	0.03 (1.40)	0.05 (1.33)	0.02 (1.32)
multiparous	−0.20 (2.59)	−0.49 (2.08)	0.06 (2.10)	0.03 (1.73)	−0.11 (1.11)	−0.07 (1.27)	−0.06 (1.23)
*p*	0.259	0.001	0.694	0.920	0.184	0.249	0.488
Smoking status							
No	−1.96 (0.71)	−0.70 (2.30)	−0.54 (1.89)	−0.96 (1.99)	0.65 (2.22)	0.14 (0.88)	−0.24 (0.83)
Yes	0.02 (2.98)	0.00 (2.42)	0.01 (2.03)	0.02 (1.70)	0.00 (1.35)	0.03 (1.32)	0.00 (1.31)
*p*	0.184	0.560	0.585	0.252	0.338	0.864	0.718
Prepregnancy BMI category							
<18.5, kg/m^2^	0.12 (4.29)	0.15 (2.76)	0.36 (1.91)	−0.07 (1.75)	0.00 (1.25)	0.03 (1.35)	−0.20 (1.11)
18.5 to <24.0, kg/m^2^	0.03 (2.67)	0.06 (2.38)	0.11 (2.08)	0.07 (1.70)	−0.03 (1.24)	0.01 (1.34)	−0.02 (1.29)
24.0 to <28.0, kg/m^2^	0.00 (2.86)	−0.22 (2.29)	−0.33 (1.95)	−0.16 (1.46)	0.13 (1.67)	0.08 (1.27)	0.15 (1.40)
≥28.0, kg/m^2^	−0.32 (2.73)	−0.20 (2.42)	−0.55 (1.78)	0.20 (2.28)	−0.05 (1.45)	0.02 (1.27)	0.04 (1.48)
*p*	0.818	0.408	0.002	0.313	0.578	0.956	0.131
Physical Activity							
<100 MET-hour/week	0.02 (3.10)	−0.18 (2.15)	−0.24 (2.09)	−0.01 (1.68)	0.10 (1.71)	0.04 (1.49)	0.00 (1.45)
100 to <200 MET-hour/week	−0.02 (3.08)	0.02 (2.41)	0.01 (1.92)	0.06 (1.69)	−0.02 (1.20)	0.06 (1.26)	−0.01 (1.25)
≥200 MET-hour/week	0.10 (2.45)	0.18 (2.75)	0.36 (2.24)	−0.07 (1.78)	−0.03 (1.24)	−0.07 (1.27)	0.03(1.26)
*p*	0.904	0.312	0.012	0.679	0.493	0.532	0.942
Calorie intake							
<2100 kcal/d	0.00 (2.74)	0.06 (2.49)	0.00 (2.06)	−0.69 (1.21)	−0.17 (1.13)	−0.21 (1.14)	−0.14 (1.06)
≥2100 kcal/d	0.04 (3.31)	−0.09 (2.30)	0.03 (1.99)	1.13 (1.78)	0.28 (1.60)	0.41 (1.49)	0.22 (1.59)
*p*	0.860	0.381	0.792	<0.001	<0.001	<0.001	<0.001
Paternal							
BMI category at enrollment							
<18.5, kg/m^2^	−0.49 (1.79)	−0.34 (2.37)	0.29 (2.01)	−0.40 (1.53)	0.43 (1.77)	−0.07(1.10)	−0.11 (0.97)
18.5 to <24.0, kg/m^2^	0.12 (3.53)	0.11 (2.38)	0.17 (1.93)	−0.06 (1.67)	0.04 (1.37)	0.06 (1.34)	0.06 (1.26)
24.0 to <28.0, kg/m^2^	−0.09 (2.51)	0.01 (2.50)	−0.07 (2.01)	0.07 (1.77)	0.01 (1.39)	−0.02 (1.34)	0.05 (1.41)
≥28.0, kg/m^2^	0.03 (2.73)	−0.25 (2.23)	−0.15 (2.16)	0.15 (1.69)	−0.11 (1.21)	0.10 (1.31)	−0.14 (1.20)
*p*	0.684	0.346	0.231	0.320	0.293	0.699	0.335
Child							
Sex							
Male	0.05 (3.03)	−0.05 (2.46)	0.01 (2.13)	−0.01 (1.77)	−0.07 (1.26)	−0.06 (1.24)	−0.09 (1.18)
Female	−0.02 (2.92)	0.05 (2.38)	0.01 (1.94)	0.04 (1.65)	0.07 (1.42)	0.11 (1.39)	0.07 (1.41)
*p*	0.710	0.545	0.976	0.624	0.098	0.064	0.064

CNY: China yuan; BMI: body mass index.

**Table 4 nutrients-13-02040-t004:** Associations of maternal dietary pattern quartiles with predicted body mass index z-scores at birth, 3, 6, 8, 12 months in the “Born in Shenyang” Cohort.

Maternal Dietary Patterns		Mean Difference in Predicted BMIZ, SD Units										
		*n*	Birth		3 Months		6 Months		8 Months		12 Months	
			β	95% CI	β	95% CI	β	95% CI	β	95% CI	β	95% CI
Food Frequency Questionnaire												
Fish–seafood pattern (FFQ)												
	Q1	234	ref	ref	ref	ref	ref	ref	ref	ref	ref	ref
	Q2	231	0.02	(−0.16, 0.19)	0.01	(−0.14, 0.16)	0.00	(−0.14, 0.15)	0.00	(−0.17, 0.16)	−0.01	(−0.22, 0.19)
	Q3	235	0.10	(−0.09, 0.28)	0.06	(−0.09, 0.22)	0.03	(−0.12, 0.18)	0.00	(−0.18, 0.17)	−0.04	(−0.25, 0.18)
	Q4	237	0.12	(−0.08, 0.31)	0.08	(−0.09, 0.25)	0.04	(−0.12, 0.20)	0.00	(−0.19, 0.19)	−0.04	(−0.27, 0.19)
Protein-rich pattern (FFQ)												
	Q1	237	ref	ref	ref	ref	ref	ref	ref	ref	ref	ref
	Q2	231	−0.20	(−0.37, −0.02)	−0.17	(−0.32, −0.03)	−0.15	(−0.30, −0.01)	−0.13	(−0.30, 0.03)	−0.11	(−0.31, 0.09)
	Q3	235	−0.17	(−0.35, 0.02)	−0.15	(−0.31, 0.01)	−0.13	(−0.29, 0.02)	−0.11	(−0.29, 0.06)	−0.10	(−0.31, 0.12)
	Q4	234	−0.16	(−0.36, 0.04)	−0.16	(−0.33, 0.01)	−0.16	(−0.33, 0.00)	−0.17	(−0.35, 0.02)	−0.17	(−0.40, 0.06)
Vegetable–fruit–rice pattern (FFQ)												
	Q1	229	ref	ref	ref	ref	ref	ref	ref	ref	ref	ref
	Q2	236	0.13	(−0.03, 0.30)	0.10	(−0.05, 0.24)	0.06	(−0.08, 0.20)	0.02	(−0.14, 0.18)	−0.02	(−0.21, 0.18)
	Q3	236	0.18	(0.01, 0.35)	0.17	(0.03, 0.31)	0.16	(0.02, 0.30)	0.15	(−0.01, 0.30)	0.13	(−0.06, 0.33)
	Q4	236	0.15	(−0.03, 0.32)	0.11	(−0.03, 0.26)	0.08	(−0.07, 0.22)	0.04	(−0.12, 0.21)	0.01	(−0.19, 0.20)
Three-day food dairies												
Traditional pattern (TFD)												
	Q1	233	ref	ref	ref	ref	ref	ref	ref	ref	ref	ref
	Q2	236	0.09	(−0.08, 0.26)	0.07	(−0.07, 0.21)	0.06	(−0.08, 0.20)	0.04	(−0.12, 0.20)	0.03	(−0.17, 0.22)
	Q3	231	0.15	(−0.03, 0.32)	0.13	(−0.03, 0.28)	0.10	(−0.05, 0.25)	0.08	(−0.09, 0.25)	0.06	(−0.14, 0.26)
	Q4	237	0.12	(−0.08, 0.32)	0.08	(−0.10, 0.25)	0.03	(−0.14, 0.20)	−0.01	(−0.20, 0.18)	−0.05	(−0.27, 0.16)
Sweet foods pattern (TFD)												
	Q1	232	ref	ref	ref	ref	ref	ref	ref	ref	ref	ref
	Q2	238	0.08	(−0.08, 0.25)	0.08	(−0.05, 0.22)	0.09	(−0.05, 0.22)	0.09	(−0.07, 0.24)	0.09	(−0.10, 0.28)
	Q3	230	0.08	(−0.09, 0.25)	0.05	(−0.09, 0.19)	0.02	(−0.12, 0.16)	−0.01	(−0.17, 0.15)	−0.04	(−0.23, 0.16)
	Q4	237	0.03	(−0.14, 0.21)	0.05	(−0.09, 0.19)	0.06	(−0.08, 0.20)	0.08	(−0.08, 0.24)	0.09	(−0.10, 0.29)
Fried food–bean–dairy pattern (TFD)												
	Q1	231	ref	ref	ref	ref	ref	ref	ref	ref	ref	ref
	Q2	230	−0.09	(−0.26, 0.08)	−0.10	(−0.25, 0.04)	−0.12	(−0.25, 0.02)	−0.13	(−0.29, 0.03)	−0.14	(−0.34, 0.05)
	Q3	233	−0.16	(−0.33, 0.01)	−0.17	(−0.31, −0.02)	−0.18	(−0.32, −0.04)	−0.19	(−0.35, −0.03)	−0.19	(−0.39, 0.00)
	Q4	243	−0.16	(−0.33, 0.01)	−0.18	(−0.33, −0.03)	−0.20	(−0.34, −0.05)	−0.22	(−0.38, −0.05)	−0.24	(−0.44, −0.04)
Whole grain–seafood pattern (TFD)												
	Q1	235	ref	ref	ref	ref	ref	ref	ref	ref	ref	ref
	Q2	237	−0.15	(−0.32, 0.02)	−0.14	(−0.28, 0.00)	−0.14	(−0.27, 0.00)	−0.13	(−0.29, 0.02)	−0.13	(−0.32, 0.06)
	Q3	231	−0.09	(−0.26, 0.08)	−0.07	(−0.21, 0.07)	−0.05	(−0.19, 0.09)	−0.04	(−0.20, 0.12)	−0.02	(−0.22, 0.18)
	Q4	234	0.01	(−0.16, 0.18)	0.03	(−0.11, 0.18)	0.06	(−0.08, 0.20)	0.09	(−0.07, 0.25)	0.12	(−0.07, 0.32)

Adjusted for other dietary patterns derived from the same dietary assessment tool (TFD or FFQ), prepregnancy BMI, age, parity, family income, education level, ethnicity, smoking status, total calorie intake, physical activity, paternal BMI. Q1: quartile 1; Q2: quartile 2; Q3: quartile 3; Q4: quartile 4; OR: odd ratio; CI: confidence interval.

**Table 5 nutrients-13-02040-t005:** Associations of maternal dietary pattern scores during pregnancy, in quartiles, with risk of being overweight/obesity across the infancy in the “Born in Shenyang” Cohort.

Maternal Dietary Patterns during Pregnancy	Risk of Being Overweight/Obesity across Infancy (ref. = Non Overweight/Obesity, *n* = 937)								
	Q1	Reference	Q2	OR	Q3	OR	Q4	OR	*p* for Trend
			(95% CI)		(95% CI)		(95% CI)		
Food Frequency Questionnaire									
Fish–seafood pattern (FFQ)									
Model 1	1.00		0.82 (0.49, 1.36)		0.98 (0.58, 1.66)		0.67 (0.40, 1.12)		0.177
Model 2	1.00		1.01 (0.58, 1.74)		1.22 (0.70, 2.12)		0.89 (0.48, 1.63)		0.623
Model 3	1.00		0.75 (0.45, 1.26)		0.93 (0.54, 1.60)		0.62 (0.36, 1.08)		0.147
Model 4	1.00		0.90 (0.52, 1.55)		1.12 (0.65, 1.92)		0.88 (0.41, 1.46)		0.435
Protein-rich pattern (FFQ)									
Model 1	1.00		0.65 (0.39, 1.08)		0.49 (0.29, 0.81)		0.71 (0.42, 1.20)		0.169
Model 2	1.00		0.62 (0.36, 1.06)		0.47 (0.26, 0.84)		0.71 (0.39, 1.31)		0.296
Model 3	1.00		0.66 (0.39, 1.13)		0.50 (0.29, 0.87)		0.75 (0.43, 1.30)		0.273
Model 4	1.00		0.65 (0.38, 1.12)		0.50 (0.27, 0.93)		0.79 (0.43, 1.47)		0.458
Vegetable–fruit–rice pattern (FFQ)									
Model 1	1.00		1.29 (0.73, 2.29)		1.62 (0.97, 2.70)		1.13 (0.65, 1.96)		0.515
Model 2	1.00		1.36 (0.76, 2.41)		1.70 (1.01, 2.87)		1.22 (0.69, 2.17)		0.468
Model 3	1.00		1.29 (0.70, 2.34)		1.73 (1.00, 2.98)		1.38 (0.77, 2.48)		0.166
Model 4	1.00		1.35 (0.74, 2.45)		1.79 (1.03, 3.12)		1.48 (0.81, 2.71)		0.166
Three-day food dairies									
Traditional pattern (TFD)									
Model 1	1.00		0.77 (0.43, 1.37)		1.11 (0.67, 1.84)		0.82 (0.48, 1.39)		0.685
Model 2	1.00		0.77 (0.43, 1.37)		1.05 (0.62, 1.79)		0.78 (0.45, 1.36)		0.633
Model 3	1.00		0.77 (0.42, 1.42)		1.18 (0.70, 1.99)		0.84 (0.44, 1.60)		0.834
Model 4	1.00		0.76 (0.42, 1.40)		1.07 (0.62, 1.86)		0.74 (0.38, 1.45)		0.552
Sweet foods pattern (TFD)									
Model 1	1.00		1.57 (0.89, 2.76)		1.13 (0.61, 2.08)		1.33 (0.74, 2.39)		0.671
Model 2	1.00		1.69 (0.97, 2.95)		1.28 (0.69, 2.36)		1.47 (0.83, 2.61)		0.683
Model 3	1.00		1.46 (0.83, 2.58)		1.02 (0.53, 1.93)		1.42 (0.76, 2.63)		0.454
Model 4	1.00		1.56 (0.89, 2.75)		1.12 (0.59, 2.16)		1.51 (0.83, 2.76)		0.663
Fried food–bean–dairy pattern (TFD)									
Model 1	1.00		0.63 (0.37, 1.05)		0.58 (0.34, 0.99)		0.66 (0.39, 1.11)		0.279
Model 2	1.00		0.62 (0.38, 1.02)		0.56 (0.33, 0.94)		0.63 (0.37, 1.07)		0.277
Model 3	1.00		0.70 (0.42, 1.16)		0.58 (0.33, 1.00)		0.65 (0.36, 1.17)		0.401
Model 4	1.00		0.68 (0.41, 1.12)		0.54 (0.31, 0.95)		0.63 (0.35, 1.14)		0.392
Whole grain–seafood pattern (TFD)									
Model 1	1.00		0.71 (0.41, 1.23)		0.79 (0.47, 1.35)		0.84 (0.50, 1.43)		0.660
Model 2	1.00		0.72 (0.40, 1.27)		0.82 (0.48, 1.38)		0.94 (0.55, 1.61)		0.651
Model 3	1.00		0.70 (0.39, 1.24)		0.72 (0.42, 1.23)		0.86 (0.50, 1.49)		0.659
Model 4	1.00		0.74 (0.40, 1.34)		0.74 (0.43, 1.28)		0.93 (0.54, 1.60)		0.501

Model 1: crude model, Model 2: adjusted for other dietary patterns derived from the same dietary assessment tool (TFD or FFQ), Model 3: adjusted for prepregnancy BMI, age, parity, family income, education level, ethnicity, smoking status, total calorie intake, physical activity, paternal BMI, Model 4: model 2 + model 3.

## Data Availability

The data presented in this study are available on request from the corresponding author.

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
