# Peer review of "Association of Maternal Dietary Patterns during Pregnancy and Offspring Weight Status across Infancy: Results from a Prospective Birth Cohort in China"

_nutrients, 2021, doi:10.3390/nu13062040_

Round 1

Reviewer 1 Report

The methodology used for the data analysis is poorly described.  It is unclear how the diet patterns were identified that accounted for 22-26% of the total variation in intake.  What accounted for the remaining variation of about 75%? 

Author Response

Response to Reviewer 1 Comments

Point 1: The methodology used for the data analysis is poorly described.  It is unclear how the diet patterns were identified that accounted for 22-26% of the total variation in intake.  What accounted for the remaining variation of about 75%?

Response 1: We apologize for a mistake in reporting the proportion of variance in the FFQ. The three dietary patterns account for 32.27%, not 22.50%, of the explained variance of total intake. We have made the corrections in the revised manuscript (page 8, line 245). We have also added more details in the methods describing how the dietary patterns were derived. Briefly, the PCA derived 21 factors which accounted for 100% of the total variation. After varimax rotation, we further identified 3 distinct factors from the FFQ and 4 factors from the three-day food diaries as dietary patterns based on eigenvalue > 1, factor interpretability after varimax rotation, and a scree plot (factors before significant inflection point of slope appears).

In our study, the identified dietary patterns explained a relatively small proportion of variation of total intake. This observation may due to the complexity of Chinese dietary patterns - many other combination patterns of food consumption were not identified as distinct dietary patterns according to the criteria. Other studies (PMID: 25821944, 30454043) focusing on Chinese dietary patterns have similarly reported small proportion of variation of the total intake from FFQs. We now include these points in the revised manuscript.

Lines 191-193: We first standardized the food consumption frequency (FFQ) and the daily food consumption level (TFD) into z-scores to reduce the influence of greater variance of food items.

Lines 195-198: Among all factors derived from FFQ and TFD, we identified distinct factors as major dietary patterns based on eigenvalue (more than 1), factor interpretability after varimax rotation, and a scree plot (factors appeared before significant inflection point of slope) (Figure S1)

Lines 244-245: Among six factors derived from the FFQ, we identified three major dietary patterns that accounted for 32.27% of the total variation (Table 2).

Lines 252-253: Among nine factors derived from the TFD, we identified four major dietary patterns that accounted for 26.31% of the total variation (Table 2).

Lines 360-365: Second, the identified dietary patterns included in the present analysis accounted for relatively small proportion of the total variation (32.27% for the FFQ and 26.31% for the TFD). This observation may due to the complexity of Chinese dietary patterns, and many other combination patterns of food consumption were not identified as distinct dietary patterns. Other studies [38,39] focusing on Chinese dietary patterns have similarly reported small proportion variation of the total intake from FFQs.

Reviewer 2 Report

The authors claim that the traditional Chinese diet is changing and is more influenced by westernized diets resulting in an increase in obesity in the young generation.  They aimed to study for the first time in China the link between maternal dietary habits (by a 3-day questionary) and its influence in offspring growth and development. The study focus was a specific population of Han ethnicity mothers with well-defined inclusion/exclusion criteria The results concluded that a diet high in proteins had a better outcome in the neonatal BMI.

Questions

1-Why was a 3-day food intake record considered a template or representative and valid characteristic to classify the mothers into the 4 group categories?

2-Why was the maternal weight gain was not recorded once the groups were determined? I think that is an important data missed in this study.

2-Was any correlation between maternal BMI and type of diet? What kind of diet did they have before the study?

3-Why did they start the study at the third trimester? Fetal development and programming of organs starts in the first and third trimester and those are the determinant windows that can be influenced by maternal dietary intake. Many studies had shown that maternal dietary composition defines offspring health in adulthood.

4-There were multiple parameters measured but there were not discussed in the results. Multiple correlations could have drawn interesting results.

5-The four groups of diets did not seem different, they even overlapped. The diets were not identified by the % of composition, named by % carbohydrates, proteins of fat.

They based the groups by patterns. The first pattern “traditional pattern (TFD)” was characterized by a high intake of tubers, vegetables, fruits, red meat and rice. The second pattern was characterized by a high intake of pastry, candy, sweet beverages, shrimps, crabs, and mussels (i.e., “sweet foods”). The third pattern was characterized by a high intake of fried foods, beans and products and dairy products (i.e., “fried food–beans-dairy” pattern), the fourth pattern was characterized by a high intake of coarse grains, shrimps, crabs and mussels and a low intake of eggs and rice (i.e., “whole grain-seafood” pattern).

7- They did not take into consideration total daily calorie intake.

6-There was no reference or definition of “high protein” diet. What was the percentage protein intake in “that diet” to be considered high in proteins? Caloric intake was not taken in consideration.

8- The claim was the influence of more westernized diet although it was not included in the 4  diets studied.

Author Response

Response to Reviewer 2 Comments

The authors claim that the traditional Chinese diet is changing and is more influenced by westernized diets resulting in an increase in obesity in the young generation.  They aimed to study for the first time in China the link between maternal dietary habits (by a 3-day questionary) and its influence in offspring growth and development. The study focus was a specific population of Han ethnicity mothers with well-defined inclusion/exclusion criteria The results concluded that a diet high in proteins had a better outcome in the neonatal BMI.

Response: Thanks for comments. We have addressed the reviewer’s concerns in our responses below.

Point 1: Why was a 3-day food intake record considered a template or representative and valid characteristic to classify the mothers into the 4 group categories?

Response 1: Both food frequency questionnaires (FFQs) and dietary records are appropriate tools to identify dietary pattens (PMID: 11790957). Although FFQs are more widely used given their advantages of feasibility and ability to capture an individual’s eating habits over long time periods, food diaries may provide more accurate assessments of intake of specific food items. In this analysis, we used both tools to derive dietary pattens during pregnancy, thus providing a more comprehensive assessment of diet. We now address this point in the revised manuscript.

Lines 69-76: Moreover, previous studies have mostly used a single dietary assessment tool to assess dietary intake, such as food frequency questionnaires (FFQ) [9,13] or food records[12]. However, both tools have their own limitations [17,18]. Although FFQs can capture an individual’s long-term habitual diet, it often overestimates food consumption and is susceptible to recall bias [19,20]. While food dairies records may provide more accurate assessment of specific food items, they usually do not reflect stable and long-term food intake [17]. The combined usage of the two tools will provide a more comprehensive assessment of diet.

Point 2: Why was the maternal weight gain was not recorded once the groups were determined? I think that is an important data missed in this study.

Response 2: We chose not to adjust for gestational weight gain (GWG) because we aimed to identify the total exposure effects of dietary patterns during pregnancy on infant growth. As GWG may potentially lie on the causal pathway between maternal diet and offspring weight status, adjusting for GWG might not only introduce bias in the association estimates, it may also over-adjust for meaningful variation in the association. Nevertheless, in the revised manuscript, we further adjusted for the GWG in a sensitivity analysis to examine whether the associations can be explained by maternal weight status. Generally, the results remained consistent. We now report these findings in Table S4.

Lines 185-187: We defined total gestational weight gain as the difference between last recorded weight before delivery and self-reported pre-pregnancy weight.

Lines 226-227: We further adjusted for total gestational weight gain to examine whether the associations can be explained by maternal weight status.

Lines 287-288: Further, adjusting for maternal gestational weight gain generally did not change the results (Table S4).

Point 3: Was any correlation between maternal BMI and type of diet? What kind of diet did they have before the study?

Response 3: In table 3, we reported the correlation between maternal BMI and dietary patterns during pregnancy. Vegetable-fruit-rice pattern (FFQ) scores were associated with pre-pregnancy weight status. Mothers with overweight or obesity had lower vegetable-fruit-rice pattern (FFQ) scores, while other dietary pattern scores were not associated with pre-pregnancy BMI category. However, we did not investigate maternal diet before pregnancy, as women in our study were enrolled at second trimester. We now include this point in the revised manuscript.

Lines 371-373: We also lacked measurements of pre-pregnancy dietary habits of mothers, thus were not able to observe the association between pre-pregnancy diet and offspring weight status.

Point 4: Why did they start the study at the third trimester? Fetal development and programming of organs starts in the first and third trimester and those are the determinant windows that can be influenced by maternal dietary intake. Many studies had shown that maternal dietary composition defines offspring health in adulthood.

Response 4: We agree with the reviewer’s point that programming of organs starts in the first trimester, thus maternal diet in early pregnancy may influence fetal development and offspring obesity. However, our study was limited by the initial time point of pregnancy health care in China’s health care system and thus, we enrolled pregnancies at second trimester (mean gestational age 22 weeks, SD 1.2). In the FFQ, we asked women to report their food intake frequency from early pregnancy to the enrolment. However, the TFDs were only conducted at middle pregnancy, thus may not adequately capture maternal diet throughout the pregnancy. We now include this point as a limitation in the revised manuscript.

Lines 367-371: Fourth, even though the FFQs collected participants’ dietary habits from early pregnancy, the TFDs were only conducted at middle pregnancy, thus may adequately capture maternal diet throughout the pregnancy. However, previous studies showed reported that dietary patterns across pregnancy are likely to remain stable across pregnancy [41].

Point 5: There were multiple parameters measured but there were not discussed in the results. Multiple correlations could have drawn interesting results.

Response 5: To examine the influence of multicollinearity of different dietary pattern scores, we created a new model in the revised manuscript (Table5, TableS5, TableS6, model 3), excluding other dietary pattern scores as confounders. Generally, the results remained stable.

Point 6: The four groups of diets did not seem different, they even overlapped. The diets were not identified by the % of composition, named by % carbohydrates, proteins of fat.

They based the groups by patterns. The first pattern “traditional pattern (TFD)” was characterized by a high intake of tubers, vegetables, fruits, red meat and rice. The second pattern was characterized by a high intake of pastry, candy, sweet beverages, shrimps, crabs, and mussels (i.e., “sweet foods”). The third pattern was characterized by a high intake of fried foods, beans and products and dairy products (i.e., “fried food–beans-dairy” pattern), the fourth pattern was characterized by a high intake of coarse grains, shrimps, crabs and mussels and a low intake of eggs and rice (i.e., “whole grain-seafood” pattern).

Response 6: We acknowledge that there were overlaps of certain food items among the dietary patterns derived from the TFD. For example, both “traditional pattern (TFD)” and “whole grain-seafood pattern (TFD)” contained rice. However, other food items from the two dietary pattens were different. The first four food items of “traditional pattern (TFD)” were tubers, vegetables, fruits and red meat, but the first few food items of “whole grain-seafood pattern (TFD)” were coarse grains, shrimps, crabs, mussels and eggs. Most parts of the principal components of the two patterns were different which makes the two patterns distinctively distinguished. Moreover, an examination of dietary patterns would parallel more closely the real world, in which nutrients and foods are consumed in combination, and their joint effects may best be investigated by considering the entire dietary pattern. This allows for food item overlaps between the dietary patterns, especially in China, a country with a particularly complex diet.

Point 7: They did not take into consideration total daily calorie intake.

Response 7: We did adjust for maternal calorie intake (total energy intake) in the fully adjusted model (Table 5, models 3 & 4). We have revised the description to total daily calorie intake to avoid ambiguity in the updated manuscript.

Lines 89-93 In the present analysis, we excluded mothers with incomplete dietary assessments during pregnancy (n= 85) or implausible values of total daily calorie intake (<500 or >3500 kcal/day) [12] (n= 138) and infants with no anthropometric data at all 5 follow up visits after birth (n= 136), leaving 937 mother-infant pairs in the analysis.

Lines 118-121We collected the hand-written TFDs before 24 weeks of gestation and summarized all food items into 21 non-overlapping food groups. We averaged each food group consumption of participants over three days and derived daily calorie and nutrient intakes according to the China Food Composition Database [22].

Lines 224-227model 3: adjusted for  pre-pregnancy BMI, age, parity, family income, education level, ethnicity, smoking status, total daily calorie intake, physical activity ,and paternal BMI;. model 4: model 2 + model 3.

Point 8: There was no reference or definition of “high protein” diet. What was the percentage protein intake in “that diet” to be considered high in proteins? Caloric intake was not taken in consideration.

Response 8: We deemed it appropriate to name the dietary pattern “protein rich pattern (FFQ)” as the main food items were all well-recognized high protein foods, such as dairy products, milk, eggs, beans and nuts. We have added a reference in the revised manuscript. However, in this analysis, we did not account for the exact nutrition intake of every dietary pattern because the aim of overall dietary pattern analysis was to detect overall food consumption habits, which would parallel more closely the real world, instead of looking at individual nutrients (PMID: 11790957). As per our response above, we did adjust for caloric intake for every dietary patten.

Point 9: The claim was the influence of more westernized diet although it was not included in the 4 diets studied.

Response 9: We now revised the introduction accordingly.

Lines 63-67To our knowledge, no previous studies have examined the association between maternal dietary patterns and offspring weight status in China, a developing country currently experiencing a rapid transition of dietary habits and also a rapid increase in prevalence of childhood obesity [1,15].

Reviewer 3 Report

The present study compares three typical dietary patterns of the Asian population, which is very interesting because most of the literature regarding this issue comes from Western countries. Mediterranean diet before and during pregnancy has shown to be the healthier dietary pattern for the mother and child, with more benefits regarding obstetric outcomes, metabolic markers in the mother and the offspring, and the best development in the early childhood and the teenager. The superiority of the Mediterranean diet, rich in vegetables, fruits, olive oil, and low in high glycemic index carbohydrates (rice) and low in saturated fats, is clear from our point of view in Europe. Therefore it is interesting to have studies based on Asian population with different patterns, for example combining vegetables, fruits and rice
. The interventions and recommendations for the mothers must take into account the traditional dietary patterns in the different countries of Asia.

The study is of remarkable methodological and scientific quality.

Author Response

Response to Reviewer 3 Comments

Point 1: The present study compares three typical dietary patterns of the Asian population, which is very interesting because most of the literature regarding this issue comes from Western countries. Mediterranean diet before and during pregnancy has shown to be the healthier dietary pattern for the mother and child, with more benefits regarding obstetric outcomes, metabolic markers in the mother and the offspring, and the best development in the early childhood and the teenager. The superiority of the Mediterranean diet, rich in vegetables, fruits, olive oil, and low in high glycemic index carbohydrates (rice) and low in saturated fats, is clear from our point of view in Europe. Therefore it is interesting to have studies based on Asian population with different patterns, for example combining vegetables, fruits and rice
. The interventions and recommendations for the mothers must take into account the traditional dietary patterns in the different countries of Asia.

The study is of remarkable methodological and scientific quality.

Response 1: We thank reviewer 3 for the positive evaluation of our study.

Reviewer 4 Report

This study aimed to investigate the association between maternal diet patterns during pregnancy and offspring weight/growth parameters at birth. The study is well justified and supported  by the literature, with a clear gap identified. The methodology is strong with excellent derivation methods for dietary patterns and good statistical analysis. 

Some abbreviations used for the first time in the abstract (OwOb) are not spelled out the first time they are used in the text, while some abbreviations in the abstract (FFQ) are spelled out the first time they appear in the text. 

Table 1 reports absolute values for BMI for the infant cohort. The WHO makes this statement about infant BMI: While the WHO growth standards include a BMI chart beginning at birth, the expert panel generally agreed that many questions about BMI during infancy remain unanswered so use of the BMI chart is not recommended for clinical use before age two years. BMI Z-scores were used for the analysis and are presented in remainer of the results section. It might be more relevant to the reader to have any understanding of the distribution of BMI Z-scores across the infant populations.

The four dietary pattens only account for 26% of the total variability--this leaves almost 75% of the variability in diet patterns unaccounted for. This appears to be a limitation that needs to be addressed. 

I am unclear on the need to include other diet patterns from the same tool in the model. What is the epidemiological justification for this? As a confounder, it would need to be associated with both the exposure and the outcome, but the intent of this analysis was to determine that exact association. Therefor they could not have been established as confounders a priori. Some additional justification for this approach would be helpful to me, otherwise the authors run the risk of over-adjustment. 

Thank you for allowing me to review this interesting paper! This is an important contribution to the literature as it describes both diet patterns and associations with important public health concerns in an understudied ethnic population. 

Author Response

Response to Reviewer 4 Comments

Point 1: This study aimed to investigate the association between maternal diet patterns during pregnancy and offspring weight/growth parameters at birth. The study is well justified and supported by the literature, with a clear gap identified. The methodology is strong with excellent derivation methods for dietary patterns and good statistical analysis.

Response 1: We thank reviewer 4 for the positive evaluation of our study. We have addressed the reviewer’s concerns in our responses below.

Point 2: Some abbreviations used for the first time in the abstract (OwOb) are not spelled out the first time they are used in the text, while some abbreviations in the abstract (FFQ) are spelled out the first time they appear in the text.

Response 2: We have revised the manuscript accordingly.

Lines 79-81: We hypothesized that maternal dietary patterns may predict infant growth trajectories and healthy dietary patterns during pregnancy were associated with decreased risk of overweight and obesity (OwOb) across infancy.

Point 3: Table 1 reports absolute values for BMI for the infant cohort. The WHO makes this statement about infant BMI: While the WHO growth standards include a BMI chart beginning at birth, the expert panel generally agreed that many questions about BMI during infancy remain unanswered so use of the BMI chart is not recommended for clinical use before age two years. BMI Z-scores were used for the analysis and are presented in remainer of the results section. It might be more relevant to the reader to have any understanding of the distribution of BMI Z-scores across the infant populations.

Response 3: A previous study has shown that the choice of WFL vs BMI to define overweight during the first 2 years of life may not greatly affect the association with cardiometabolic outcomes during early adolescence. In this regard, BMI can be used as a reliable growth metric for epidemiologic research when examining the weight status of children younger than 2 years (PMID: 30646168). Moreover, we conducted additional sensitivity analysis using WFLZ as an indicator for infant weight status to assess the robustness of the results. Generally, the results remained consistent. We now report these findings in the revised manuscript and in Table S3 and Table S6. We relaced BMI with BMIZ to describe infant weight status in Table 1.

Lines 218-219: In the sensitivity analysis, we also used WFLZ as an indicator for infant weight status to assess the robustness of the results.

Lines 285-287: In sensitivity analyses, results remained stable when using WFLZ as an indicator of infant weight status instead of BMIZ (Table S3).

Lines 288-290: We also observed higher “protein rich pattern (FFQ)” score quartiles and “fried food-bean-dairy pattern (TFD)” score quartiles were associated with lower infant BMIZ (Table S5) and WFLZ (Table S6) across infancy.

Point 4: The four dietary pattens only account for 26% of the total variability--this leaves almost 75% of the variability in diet patterns unaccounted for. This appears to be a limitation that needs to be addressed.

Response 4: Please see our response to the same comment by Reviewer #1 above, on page 1 of this letter. 

Point 5: I am unclear on the need to include other diet patterns from the same tool in the model. What is the epidemiological justification for this? As a confounder, it would need to be associated with both the exposure and the outcome, but the intent of this analysis was to determine that exact association. Therefor they could not have been established as confounders a priori. Some additional justification for this approach would be helpful to me, otherwise the authors run the risk of over-adjustment.

Response 5: We adjusted for other dietary patterns in our regression models, as different dietary patterns may not only be associated with each other, but also potentially associated with offspring weight status. We added a model to check the robustness of the results when excluding other dietary pattern scores as confounders. Generally, the results remained consistent. We now report these findings in the revised manuscript and in Table 5, TableS5 and Table S6.

Point 6: Thank you for allowing me to review this interesting paper! This is an important contribution to the literature as it describes both diet patterns and associations with important public health concerns in an understudied ethnic population.

Response 6: Thanks for all the comments.

Round 2

Reviewer 2 Report

Thank you for responding and clarifying my comments. 

Author Response

Thanks for the comments.